# Diagnostic Biomarkers in Renal Cell Tumors According to the Latest WHO Classification: A Focus on Selected New Entities

**DOI:** 10.3390/cancers16101856

**Published:** 2024-05-13

**Authors:** Francesca Sanguedolce, Roberta Mazzucchelli, Ugo Giovanni Falagario, Angelo Cormio, Magda Zanelli, Andrea Palicelli, Maurizio Zizzo, Albino Eccher, Matteo Brunelli, Andrea Benedetto Galosi, Giuseppe Carrieri, Luigi Cormio

**Affiliations:** 1Pathology Unit, Policlinico Foggia, University of Foggia, 71122 Foggia, Italy; francesca.sanguedolce@unifg.it; 2Section of Pathological Anatomy, Department of Biomedical Sciences and Public Health, United Hospitals, Università Politecnica delle Marche, 60126 Ancona, Italy; r.mazzucchelli@univpm.it; 3Department of Urology and Renal Transplantation, Policlinico Foggia, University of Foggia, 71122 Foggia, Italygiuseppe.carrieri@unifg.it (G.C.); luigi.cormio@unifg.it (L.C.); 4Department of Molecular Medicine and Surgery, Karolinska Institutet, 17177 Stockholm, Sweden; 5Department of Urology, Azienda Ospedaliero-Universitaria Ospedali Riuniti Di Ancona, Università Politecnica Delle Marche, Via Conca 71, 60126 Ancona, Italy; 6Pathology Unit, Azienda USL-IRCCS di Reggio Emilia, 42123 Reggio Emilia, Italy; magda.zanelli@ausl.re.it (M.Z.); andrea.palicelli@ausl.re.it (A.P.); 7Surgical Oncology Unit, Azienda USL-IRCCS di Reggio Emilia, 42123 Reggio Emilia, Italy; maurizio.zizzo@ausl.re.it; 8Department of Pathology and Diagnostics, University and Hospital Trust of Verona, 37126 Verona, Italy; albino.eccher@azosp.vr.it; 9Department of Pathology and Diagnostics and Public Health, Section of Pathology, University Hospital of Verona, 37126 Verona, Italy; matteo.brunelli@univr.it; 10Department of Urology, Bonomo Teaching Hospital, 76123 Andria, Italy

**Keywords:** WHO classification, renal cell carcinoma, immunohistochemistry, diagnostic markers, urogenital tumors

## Abstract

**Simple Summary:**

The latest World Health Organization classification for kidney tumors, released in 2022, brings important changes, introducing new types of tumors and refining the names of certain diseases. This classification highlights the need for both traditional and modern methods in diagnosing these tumors, including molecular and genetic analysis alongside conventional microscopic examination. Our article focuses on the role of specific markers identified through immunohistochemistry, a technique used to study proteins in tissues, in diagnosing these new tumors. We critically discuss the features of these tumors, such as the clear cell capillary renal cell tumor, eosinophilic solid and cystic renal cell carcinoma, and other oncocytic tumors. By understanding these markers and their significance, pathologists can better diagnose these tumors, leading to improved treatment strategies and outcomes for patients. Our review provides valuable insights into these novel tumors, offering guidance for healthcare professionals in their daily practice, ultimately benefiting society by enhancing diagnostic accuracy and patient care.

**Abstract:**

The fifth edition of the World Health Organization (WHO) classification for urogenital tumors, released in 2022, introduces some novelties in the chapter on renal epithelial tumors compared to the previous 2016 classification. Significant changes include the recognition of new disease entities and adjustments in the nomenclature for certain pathologies. Notably, each tumor entity now includes minimum essential and desirable criteria for reliable diagnosis. This classification highlights the importance of biological and molecular characterization alongside traditional cytological and architectural features. In this view, immunophenotyping through immunohistochemistry (IHC) plays a crucial role in bridging morphology and genetics. This article aims to present and discuss the role of key immunohistochemical markers that support the diagnosis of new entities recognized in the WHO classification, focusing on critical topics associated with single markers, in the context of specific tumors, such as the clear cell capillary renal cell tumor (CCPRCT), eosinophilic solid and cystic renal cell carcinoma (ESC-RCC), and so-called “other oncocytic tumors”, namely the eosinophilic vacuolated tumor (EVT) and low-grade oncocytic tumor (LOT). Their distinctive characteristics and immunophenotypic profiles, along with insights regarding diagnostic challenges and the differential diagnosis of these tumors, are provided. This state-of-the-art review offers valuable insights in biomarkers associated with novel renal tumors, as well as a tool to implement diagnostic strategies in routine practice.

## 1. Introduction

The fifth edition of the World Health Organization (WHO) classification for urogenital tumors, commonly referred to as the “Blue Book” and released in 2022, provides peculiar revisions in the field of renal cell carcinoma (RCC) compared to the previous 2016 classification [1].

The most significant changes include the introduction of new disease entities, as well as changes in the nomenclature for some pathologies (see Table 1). Another distinctive feature of the fifth edition of the WHO classification is the introduction, for each tumor entity, of minimum essential and desirable criteria that enable a reliable diagnosis of them [2].

Overall, this new classification provides a more precise distinction of the various subgroups of tumors and emphasizes the importance of their biological and molecular characterization alongside traditional cytological and architectural features. In this setting, immunophenotyping (i.e., the identification of tissue biomarkers through immunohistochemistry (IHC)) plays an even greater role than in the past, as it represents a bridge between morphology and genetics through the determination of diagnostic protein biomarkers identifiable in tissue sections [3]. Since there is no specific immunophenotypic biomarker for a particular histotype of renal tumor, it is essential to apply a panel of markers that, based on morphological characteristics, allow us to favor or rule out the diagnostic hypotheses formulated on a case-by-case basis.

The aim of this paper is to present and discuss the role of the main immunohistochemical markers that often provide substantial support to the diagnosis of the new entities recognized in the current WHO classification, with a specific focus on critically analyzing the advantages and disadvantages associated with the use of individual markers.

## 2. Clear Cell Papillary Renal Cell Tumor (CCPRCT)

The CCPRCT was initially identified as a tumor associated with the end-stage renal disease setting and only later found in sporadic cases [4,5]. The initial designation of carcinoma was due to its significant morphological similarities to clear cell renal cell carcinoma (CCRCC), and for this reason it may be diagnostically challenging in routine practice [6]. Nevertheless, the identification of the lack of von Hippel Lindau (VHL) gene alterations, chromosome 3p35 loss, and trisomy 7/17, along with follow-up data indicating an indolent clinical behavior, with no evidence of any recurrence or metastasis so far, supported the name change from “carcinoma” to “tumor” in the new classification [7,8]. Xu et al. highlighted its genomic stability and distinct metabolic phenotype, with severe depletion of mitochondrial DNA resulting in oxidative stress [9]; such findings further highlight the molecular-level distinction between the CCPRCT and more common and morphologically similar renal epithelial malignancies, CCRCC, and papillary renal cell carcinoma (PRCC).

Since the correct diagnosis of the CCPRCT results in complete tumor excision being curative, with no need of any further treatment, its accurate diagnosis is of pivotal importance.

At gross examination, these are small (pT1), cortical, well-circumscribed lesions, which can be multiple in as many as one-fifth of all patients [8]. Microscopically, low-grade (1–2 WHO/ISUP) clear cells with suprabasal nuclei in a linear array above the basement membrane are arranged in a variable mixture of cystic, branched, acinar and tubular, solid, papillary formations. Lack of tumor necrosis or perineural and lymphovascular invasion is a constant finding.

### 2.1. Immunophenotype

Positivity for CK7 (diffuse staining), CAIX (“cup-shaped” expression, with staining of the basolateral but not the luminal portion of tumor cells), GATA3, PAX2, PAX8, 34βE12, AE1/AE3, or CAM5.2.

Negativity for AMACR, CD10, RCCm, or cathepsin K [4,5,10,11] (Figure 1).

### 2.2. Critical Topics and Insights

The assessment of CAIX, CK7, and CD10 immunostaining is regarded as a desirable diagnostic criterion in these tumors, and therefore it is advisable to apply a minimal panel consisting of these three biomarkers in routine practice, to be expanded both in challenging cases [8] and in those rare instances where combined features of CCRCC and the CCPRCT may occur [12]. In the latter case, it has been recommended to label the tumor as CCRCC if the immunophenotype does not perfectly match. The possible presence of “mixed” tumors might affect a straightforward diagnosis of the CCPRCT, especially in the biopsy setting [12]. Notably, according to the results from a recent ISUP consensus conference, IHC was used as a confirmatory tool even in morphologically typical cases by 77.4% of respondents [12], with molecular studies being advocated as second-line diagnostic tests, to be applied only in equivocal cases.

The “cup-shaped” CAIX staining is considered typical of the CCPRCT, unlike the complete membranous pattern of labeling in CCRCC. Nevertheless, such basolateral CAIX expression has been described, though occasionally, in CCRCC as well, especially in pseudopapillary areas, which are morphological mimickers of the CCPRCT [13].

Recently, CD10 and AMACR positivity has been described in as many as 53% and 35% of CCPRCT cases, respectively, in a study on cytology specimens [14]. Furthermore, AMACR has been recently reported as focally positive in cystic spaces or papillary fronds of the CCPRCT [13].

The frequent expression of 34βE12 and GATA3 suggests that these tumors, unlike CCRCC, may derive from the distal rather than proximal nephron, despite their clear cell morphology [15]. GATA3 staining raises the diagnostic possibility of a urothelial carcinoma (UC), especially upon renal biopsy; moreover, UCs of the upper urinary tract may be PAX8-positive in approximately 20% of cases [16].

The main differential diagnosis of the CCPRCT is with other renal tumors which may exhibit low-grade, clear cell, and/or papillary morphology, such as CCRCC and PRCC (see Table 2). In a broader context, the very rare ELOC-mutated RCC has overlapping morphological features with the CCPRCT, namely low stage and grade, the presence of branching tubules and papillae, and clear cytoplasm. Like the CCPRCT, ELOC-mutated RCC is CK7/CAIX/CK34βE12-positive, and the distinction between these two entities relies on a lack of prominent subnuclear vacuolization and the presence of CD10 staining in the latter [17].

## 3. Eosinophilic Solid and Cystic Renal Cell Carcinoma (ESC-RCC)

In the current WHO classification, eosinophilic solid and cystic renal cell carcinoma (ESC-RCC) has been introduced as a new entity, under the category “Other renal tumours”. These cortical tumors are mainly pT1, well-circumscribed, sporadic, and solitary lesions, often incidentally diagnosed [18,19]. Microscopically, they show a solid and cystic appearance, and tumor cells in solid areas have abundant eosinophilic cytoplasm, whereas hobnail cells line the cystic spaces. Under electron microscopy, characteristic aggregates of rough endoplasmic reticulum and granular material are seen in tumor cells, appearing as basophilic to dark, coarse, intracytoplasmic granules (stippling) under light microscopy. Other morphological findings are psammoma bodies and intracytoplasmic vacuoles [20].

Typically, biallelic losses or mutations in the TSC1/TSC2 genes are found in ESC-RCC, resulting in activation of mTOR complex 1; only few cases have been described in patients with tuberous sclerosis complex (TSC) [21,22]. Due to its peculiar molecular setting, a correct diagnosis of ESC-RCC is of pivotal importance for patient selection in targeted therapy with mTOR inhibitors [23].

ESC-RCC has a benign clinical behavior in most cases, with few metastatic cases being reported so far [23,24]; such unfavorable behavior was associated with larger size and morphological aggressive features, namely necrosis and hemorrhage [23,24,25].

### 3.1. Immunophenotype

Positivity for cathepsin k (usually patchy, focal to diffuse), CK20 (patchy, focal to diffuse, may be negative in up to 20% of cases), PAX8, AE1/AE3 (rarely focal to negative), CK8/18, vimentin, or melan-A (occasionally).

Negativity for CK7 (rarely focal expression), CD117(KIT), CAIX, or HMB45 [19,23,26,27].

### 3.2. Critical Topics and Insights

While the typical constellation of morphological features is an essential diagnostic criterion of ESC-RCC, CK20 immunoreactivity along with mutations in TSC1 or TSC2 genes are listed among desirable criteria [18].

The combined CK20pos/CK7neg staining in ESC-RCC is unique among RCC subtypes [26]. Furthermore, even the uncommon CK20-negative cases are not simultaneously CK7-positive, allowing for a distinction from more frequent oncocytic renal tumors, such as chromophobe RCC (ChrRCC) [28].

Li et al. reported that approximately 30% of cases from a series of 33 patients aged 35 years or younger diagnosed with “unclassified” RCCs with a predominantly eosinophilic cytoplasm were reclassified as ESC-RCCs, thus prompting the need to perform CK20 IHC in young patients with oncocytic tumors [29]. Notably, Akgul et al. recently investigated the immunophenotypes of 300 cases of RCC that were unclassified, finding CK20 staining in 5 cases, which did not show morphologic features of ESC-RCC, including cystic growth or coarse intracytoplasmic granules [30].

All in all, the peculiar combination of positivity for CK20 and cathepsin K as well as the lack of staining for CK7 and CD117 allows us to reliably diagnose ESC-RCC in the proper clinical and morphological setting. Since it is a still poorly known histotype, especially in a sporadic setting, some of its biological and clinical features are yet to be defined, which will likely occur only over time.

## 4. Oncocytic Tumors

The current classification of renal epithelial tumors introduces a category of “Oncocytic and chromophobe renal cell tumors”, which encompasses two well-known diagnostic entities, namely renal oncocytoma (RO) and ChrRCC, along with two emerging entities (collectively referred to as “Other oncocytic tumors of the kidney”), namely the low-grade oncocytic tumor (LOT) and the eosinophilic vacuolated tumor (EVT). The latter two tumors exhibit several overlapping features: the presence of eosinophilic cells, indolent behavior, and distinct molecular alterations, such as inactivating somatic mutations of the TSC2 gene or activating mutations of mTOR, as seen in ESC-RCC [31,32]. The presence of an overlapping genetic profile in these three tumors might lead to the proposal of grouping them separately into the same category, but the significant morphological and immunohistochemical differences among them, as well as the fact that such molecular alterations have been identified in other renal tumors, do not support this hypothesis [15].

As for now, the differential diagnosis between these two tumors does not have significant implications in clinical practice [33]. Nevertheless, since they are emerging entities, which, most probably, have likely been misdiagnosed in the past as other histotypes of oncocytic tumors or as unclassified RCCs, we cannot reliably determine their incidence or biological behavior, at least until these tumors have fully entered routine diagnostics and can be the subject of further clinical and molecular studies. All in all, the aim of this new classification is to keep these two tumors in a separate subgroup from tumors labeled as “Renal cell carcinoma NOS”, in order to reduce the latter wastebasket category for use with highly aggressive tumors only [1].

From the practicing pathologist’s point of view, the distinction among entities in the spectrum of oncocytic tumors may be challenging when handling situations with limited tissue in needle biopsy specimens. In a recent state-of-the-art review of the LOT [34], a protocol was suggested in this setting, including meticulous evaluation of morphology alongside the immunohistochemical profile (see Table 3), in order to achieve the most accurate diagnosis possible. However, in cases where this approach may not be feasible, the pathology report will include a descriptive diagnosis like “oncocytic tumor, not further specified”, accompanied by a comment detailing the list of potential differential diagnoses and/or preferences [34].

### 4.1. Eosinophilic Vacuolated Tumor (EVT)

The eosinophilic vacuolated tumor (EVT), previously known as the high-grade oncocytic tumor [35] or RCC with eosinophilic vacuolated cytoplasm [36], is an uncommon entity, with more than 50 cases reported to date, all exhibiting a benign behavior, with no evidence of recurrences or metastases [19,28,32,35,37,38].

The EVT is a solitary and sporadic tumor, only rarely presenting in patients with TSC, and it is not larger than 7 cm (mean tumor size between 3 and 4 cm) [19,28,32,35,36,39,40]. Under light microscopy, tumor cells have an abundant eosinophilic cytoplasm with prominent intracytoplasmic vacuoles, and round to oval nuclei with striking, sometimes large nucleoli, falling into the category of nuclear grade WHO/ISUP 3, and peripheral thick-walled vessels are a common finding [35,36,38]. The oncocytic appearance of the cytoplasm is due to an increased number of intracytoplasmic mitochondria, as with an oncocytoma and the LOT, as highlighted by ultrastructural analysis [28].

Mutations in the TSC/MTOR pathway leading to mTORC1 activation have been consistently reported in the EVT, with a recent study describing non-overlapping mutations in MTOR, TSC2, and TSC1 in all cases, alongside with low mutational rates [38]. Limited data exist on the occurrence of losses of chromosomes 1 and 19p in this tumor [32].

#### 4.1.1. Immunophenotype

Positivity for CD117 (KIT), CD10, cathepsin K (sometimes focally), CKs, PAX8, FH, or SDHB

Negativity for CK7 (can be positive only in rare, scattered cells), CK20, HMB45, Melan A, vimentin (can be positive only in rare, scattered cells), or TFE3 [19,35,38].

#### 4.1.2. Critical Topics and Insights

EVT cells are mostly CD117-positive, and they are negative or positive only in rare cells for CK7 and, likewise, RO. However, the distinction between these two tumors is supported by (1) the different expression of cathepsin K (which is often positive in the EVT, while usually negative in RO) and CD10 (present in the EVT, absent in RO) and (2) the presence of widespread, high-grade morphological features, which rules out the diagnosis of RO [15,28,41].

The differential diagnosis between the EVT and ChrRCC is easier to make, in that it relies on all the previous markers plus CK7, which is typically, diffusely, and strongly expressed in ChrRCC; furthermore, the latter tumor lacks prominent cytoplasmic vacuoles and striking nucleoli, whereas the EVT does not feature typical ChrRCC raisinoid nuclei with perinuclear halos [28,31]. Interestingly, as many as 2% of eosinophilic tumors from the series of 300 unclassified RCC reviewed by Akgul et al. showed both morphological (high nuclear grade, intracytoplasmic vacuoles) and immunophenotypical (diffuse cathepsin K expression) features consistent with those of the EVT [30].

CD117 expression may be variable in the EVT, though present in the majority of cases [19]. The uncommon CD117-negative cases enter a differential diagnosis of angiomyolipomas (AMLs), which are CK7-negative and cathepsin K-positive, prompting the need for adding further markers (cytokeratins, PAX8, or both), especially in a biopsy setting.

### 4.2. Low-Grade Oncocytic Tumor (LOT)

The low-grade oncocytic tumor (LOT) is typically a single, sporadic tumor with intermediate features between RO and ChrRCC [15,19,32,41,42]. It can be multifocal in the end-stage disease and TSC setting [39,43,44,45]. According to reports so far, its clinical behavior is indolent [42,43,44,45,46]. At a molecular level, from the limited data available to date, the LOT seems to share some features with RO such as the loss of 1p36 and a diploid pattern. The common involvement of the mTOR pathway genes has been demonstrated in this tumor, further supported by the expression of the mTOR pathway activation markers p-S6 and p-4EBP1 [37,44,45,47]; conversely, these tumors lack complete chromosomal gains or losses, as well as CCND1 rearrangements [43].

Grossly, the LOT is a smaller, solid tumor with a mahogany-brown to tan cut surface (similar to that of RO). Under light microscopy, the most common growth patterns are solid and compact-nested, focal tubular, tubulo-reticular, or trabecular [41,42,43,46]. The tumor cells are eosinophilic due to the presence of abundant, closely packed cytoplasmic mitochondria, similar to those of an oncocytoma and the EVT, which can be identified under electron microscopy [28,40]. Nuclei are bland, round to oval, sometimes showing perinuclear clearing (halos), with Muller–Mowry colloidal iron staining being either negative or only luminal-positive. Oftentimes, sharply delineated, edematous, and hemorrhagic areas with loose cell growth (“boats in a bay”) can be seen [19,42]. Recently, Gupta et al. raised the issue of a tumor with morphologic overlap between the LOT and eosinophilic variant of ChrRCC, including prominent raisinoid nuclei and occasional binucleation, while both its immunoprofile and clinical behavior were consistent with those of the former [48]. The LOT lacks aggressive morphological features, such as necrosis, severe nuclear atypia, and increased mitotic activity.

#### 4.2.1. Immunophenotype

Positivity for CK7 (strong and diffuse), GATA3, p-S6 and p-4EBP1 (at least focally), AE1/AE3, PAX8, E-cadherin, BerEP4, MOC31, FH, or SDHB.

Negativity for CD117, CAIX, CK20, CK5/6, p63, CD15, HMB45, melan-A, vimentin (may be positive), CD10 (may be focally positive), AMACR (may be focally positive), or cathepsin K [32,42,43,45,47] (Figure 2).

#### 4.2.2. Critical Topics and Insights

While some authors suggested that typical morphological features alone allow for the identification of the LOT under light microscopy, others advocate for the use of a 2-marker antibody panel consisting of CD117 and CK7 to reliably differentiate this tumor from other oncocytic tumors, mainly RO and ChrRCC [49]. Nevertheless, there has been a recent suggestion that a distinct entity, referred to as “oncocytic renal neoplasm with diffuse keratin 7 immunohistochemistry”, shows the same immunoprofile along with RO-like morphology, and its connection with the LOT is still debatable [50].

The immunophenotype of the LOT has been investigated in two homogeneous series of 28 and 23 tumors, respectively, by Trpkov et al. [42] and Akgul et al. [46]. All reported cases were CK7-positive, whereas CD117 staining was found in 7 out of 28 tumors in one series only [42]. Furthermore, vimentin expression was diffusely present in most of the positive cases, accounting for 35% of the cases described by Akgul et al. [46]. Notably, approximately 73% of ROs from a large series were reported as vimentin-positive, with stained cells mostly at the edge of a central scar or scattered throughout [51], thus supporting its ineffectiveness as a marker in the differential diagnosis of oncocytic tumors.

GATA3 has been reported as a consistently positive marker in the LOT [47,52]. GATA3 staining, though focal and variable, has been reported in ChrRCC, as well as in a small subset of ROs from a pan-tumor immunohistochemical study [53]; nevertheless, since this study dates back to 10 years ago, it has been suggested that at least some of the cases diagnosed as ChrRCC and RO at that time, if reviewed today, would be reclassified into other diagnostic categories within the spectrum of oncocytic tumors, including the LOT [34].

## 5. Immunohistochemical Biomarkers in Selected Renal Tumors: A Primer

The main markers discussed in this review are summarized in Table 4.

Carbonic anhydrase IX (CAIX) is a transmembrane isozyme belonging to the carbonic anhydrase family involved in the regulation of intra- and extracellular pH, cell proliferation and adhesion, and the transfer of CO2 across the renal tubules. It is regulated by hypoxia-inducible factor (HIF) and overexpressed in the setting of tissue hypoxia and acidic tumor microenvironment [54]. Unlike other members of its family, CAIX is suppressed in healthy kidneys by the wild-type von Hippel-Lindau (VHL) protein; hence, normal renal cells are CAIX-negative. Conversely, CAIX is expressed in the vast majority of CRCCs through ubiquination-independent HIF-1α and 2α accumulation due to hypoxia or inactivating mutations in the VHL tumor suppressor gene [55,56].

CAIX strong and diffuse staining is a constant finding in clear cell renal tumors, namely CCRCC and CCRPT. In keeping with that, most tumors with clear cell morphology from a series of 300 unclassified RCC were CAIX-positive [30]. Recently, Baniak et al. confirmed its role as a highly sensitive diagnostic marker in CCRCC and CCPT, while highlighting its low specificity across the whole spectrum of renal epithelial neoplasms (except for RO and ESC-RCC) [57].

A variable degree of CAIX staining can be seen in several nonrenal tumors, including neuroendocrine tumors, mesotheliomas, and tumors of the endometrium, stomach, cervix, breast, lung, liver, testis, and brain, which is a pitfall in distinguishing renal versus nonrenal carcinomas [3].

Viable cells adjacent at the edge of tumor necrosis due to ischemia and hypoxia show non-specific CAIX expression [58].

Cytokeratins (CKs) are intermediate filaments that allow for the identification of epithelial cells; CKs of different molecular weight are characteristic of different epithelial types. In renal tumors, the most common CK is CK7, a low-molecular-weight CK which is highly expressed in most papillary renal cell carcinomas (PRCCs), ChrRCCs, collecting duct carcinomas (CDCs), and almost all CPRCTs and mucinous tubular and spindle cell carcinomas (MTSCCs) [59]. Conversely, CK7 staining is a rare and focal finding in CCRCC and tubulocystic RCC, and it is very focal to almost absent in RO [3]. CK7 is also a marker of urothelial carcinoma; therefore, an upper tract urothelial carcinoma should be always taken into account when assessing a CK7-positive renal neoplasm.

Urothelial carcinoma cells express other CKs, namely CK20 and 34βE12 as well, which are low-molecular-weight and high-molecular-weight CKs, respectively. However, CK20 expression is lacking in the vast majority of renal cell tumors, except ESC-RCC, whereas 34βE12 is a marker of CDC, renal medullary carcinoma (RMC), and most CCPRCTs [3].

CD10 (also known as acute lymphocytic leukemia antigen or CALLA) is a cell-surface glycoprotein expressed on the brush border of the proximal renal tubular epithelial cells and podocytes. It has been traditionally regarded as a marker of CRCC, PRCC, and CDC [3]. Other renal tumors can at least partly express CD10, such as ChrRCC and RO [56]. In the study by Akgul et al. on a series of unclassified RCCs, CD10 was one of the most common markers expressed by tumors with both clear cell RCC-like (92%) and papillary RCC-like/collecting duct carcinoma-like/pure sarcomatoid morphologies (82%) [30]. It is an overall highly sensitive yet non-specific marker for renal tumors, in that its expression can be found in several nonrenal tumors, including carcinomas of the lung, colon, pancreas, ovary, and bladder, as well as soft tissue tumors, melanomas, and lymphomas [60].

CD117 (C-KIT or KIT) is a transmembrane tyrosine kinase receptor encoded by a proto-oncogene that, upon binding to its ligands, regulates cell survival, proliferation, and differentiation.

Several tissues throughout the body show CD117 expression, including hematopoietic precursors, mast cells, and melanocytes, as well as tumors arising from these cells [56].

The combined application of CK7 and CD117 is a reliable first-level panel to discriminate lesions in the spectrum of eosinophilic/oncocytic tumors, with RO being CK7neg/CD117pos, ChrRCC CK7pos/CD117pos, the EVT CK7neg/CD117pos, the LOT CK7pos/CD117neg, and ESC-RCC CK7neg/CD117neg [59] (see Table 3). Sarcomatoid RCC, angiomyolipoma, and pelvic urothelial carcinoma may variably stain for CD117 [54].

Tumors expressing CD117 are susceptible to therapy with tyrosine kinase inhibitors (TKIs), such as imatinib mesylate, which is currently the basis for treatment of gastrointestinal stromal tumors (GISTs), due to the presence of underlying mutations in exons 9 and 11 of the C-KIT gene; however, such mutations have not been identified in CD117-positive renal tumors, thus undermining the efficacy of TKIs in this setting [54,56].

Cathepsin K, a member of the papain-like cysteine peptidase family, is primarily involved in bone resorption but has broader implications in various physiological processes and diseases, including cancer, since its expression and activity are linked to tumor growth, metastasis, and cancer cell invasion [61]. Though not a marker of common histotypes of renal epithelial tumors, it is frequently detected in a subset of renal tumors, as well as in perivascular epithelioid cell tumor (PEComa)/AML [59]. Specifically, among RCCs with a clear cell and papillary architecture, cathepsin K is essential for distinguishing translocation RCC from the more common CCPRCT, which typically lacks expression of this marker. Among renal tumors characterized by eosinophilic cell morphology, cathepsin K expression is observed in ESC-RCC and translocation RCC [3]. Conversely, common eosinophilic/oncocytic renal tumors such as RO, the ChrRCC eosinophilic variant, and oncocytic PRCC are typically negative for cathepsin K, although some variability has been observed in recent studies [62]. Notably, the capillaries and associated macrophages are cathepsin K-positive, thus serving as an internal control for evaluating staining quality.

α-Methylacyl coenzyme A racemase (AMACR) is a mitochondrial enzyme which mediates fatty acid oxidation, and is typically present in normal hepatocytes, proximal renal tubules, and bronchial cells. AMACR is far from being a lineage specific marker: prostatic tumor cells commonly express this AMACR, as well as adenocarcinomas arising from the liver, bladder, colon, ovary, breast [60]. The vast majority of PRCCs are AMACR-positive, along with MTSCRCC and translocation RCC. AMACR expression has been described in up to 78% of CCRCCs according to a recent study, mostly high-grade [63]. Conventional chromophobe/oncocytic renal neoplasms lack this marker.

GATA3 (GATA binding protein 3 to DNA sequence [A/T] GATA[A/G]) is a transcription factor which plays a critical role in stimulating cell proliferation, development, and differentiation. It is regarded as a sensitive, though not specific, marker of urothelial and mammary lineage, in that its expression has been described, at least partly, in several tumor types [3,53]. In the spectrum of renal epithelial tumors, GATA3 is usually expressed by PRNRP and CCPRCT, while being consistently negative in other tumor types.

## 6. Conclusions

The evolving spectrum of renal tumors, including novel and emerging entities according to the current WHO classification, poses challenges in everyday diagnostic practice. While viewing hematoxylin and eosin-stained slides is still the basis to reliably identify these lesions, IHC strengthens its role as a valuable tool for accurate diagnosis, especially when managing small biopsy specimens; a panel of IHC markers is often needed by the pathologist to support and refine morphological evaluation. In this setting, the pathologist should acknowledge issues and pitfalls of IHC assessment, as well as the specific features of each marker in the proper histopathological context.

The ongoing changes in renal tumor classification are constantly driven by advances in molecular pathology and immunohistochemical techniques, resulting in the definition of entities with more and more precise clinical and prognostic characteristics. The integration of various methodologies and the ongoing search for biomarkers with increasing specificity and sensitivity for individual diagnostic entities are of crucial importance for the accurate management of patients.

## Figures and Tables

**Figure 1 cancers-16-01856-f001:**
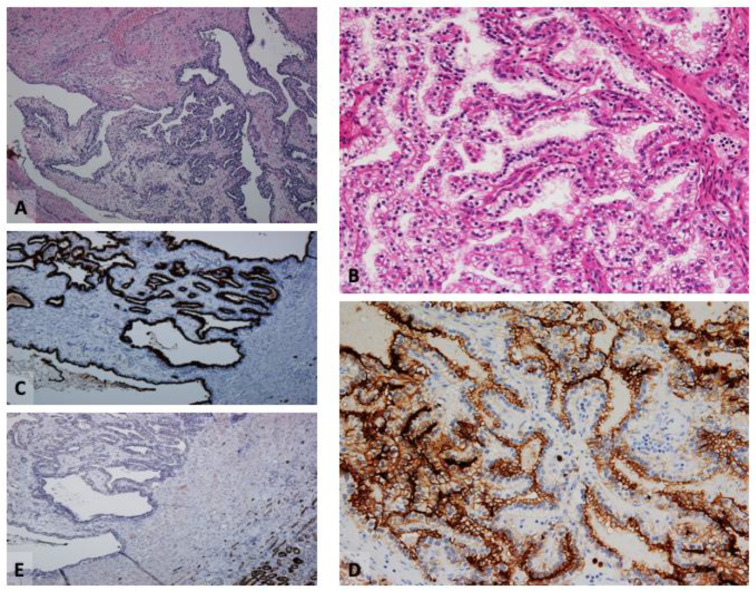
Clear cell papillary renal cell tumor. (**A**,**B**) Hematoxylin–eosin (H&E)-stained section showing tumor cells with characteristic suprabasal nuclei, arranged in branched, sometimes cystic, tubular formations (original magnifications ×100 (**A**) and ×200 (**B**)). (**C**,**D**) Tumor cells are positive for CK7 (original magnification ×100 (**C**)), and CAIX, with the typical cup-shaped staining pattern (original magnification ×200 (**D**)). (**E**) Tumor cells are negative for AMACR (original magnification ×100).

**Figure 2 cancers-16-01856-f002:**
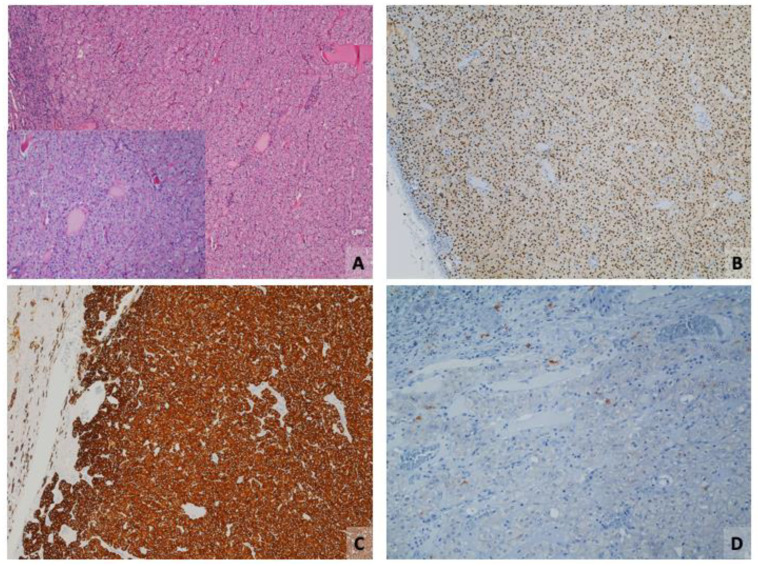
Low-grade oncocytic tumor. (**A**) Hematoxylin–eosin (H&E)-stained section showing eosinophilic tumor cells, often with nuclear halos (inset) (original magnifications ×100 and ×200 (inset)). (**B**,**C**) Tumor cells are diffusely positive for PAX8 ((**B**) original magnification ×100), and CK7 ((**C**) original magnification ×100). (**D**) Tumor cells are negative for CD117 (original magnification ×100); scattered positive mast cells.

**Table 1 cancers-16-01856-t001:** Brief summary of the changes in the fifth edition of the WHO classification of renal tumors which are discussed in this review, along with the underlying reason.

WHO 2022	WHO 2016	Reason
Clear cell papillary renal cell tumor (CCPRCT)	Clear cell papillary renal cell carcinoma (CCPRCC)	Benign behavior
Eosinophilic solid and cystic renal cell carcinoma (ESC-RCC)	Not present	New entity in the subgroup of “Other renal tumours”
Low-grade oncocytic tumor (LOT)	Not present	Emerging entity in the subgroup of “Other oncocytic tumours of the kidney”
Eosinophilic vacuolated tumor (EVT)	Not present	Emerging entity in the subgroup of “Other oncocytic tumours of the kidney”

**Table 2 cancers-16-01856-t002:** Differential diagnosis of low-grade tumors with clear cell and/or papillary features. CCPRCT: clear cell papillary renal cell tumor; CCRCC: clear cell renal cell carcinoma; PRCC: papillary renal cell carcinoma.

Markers	CCRCC		CCPRCT		PRCC
CAIX	+	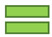	+	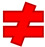	−
CK7	−/+	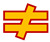	+	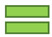	+
AMACR	−/+	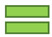	−/+	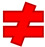	+
CD10	+	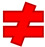	−/+	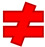	+
34βE12	−	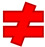	+/−	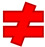	−
GATA3	−	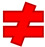	+	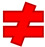	−

**Table 3 cancers-16-01856-t003:** Immunohistochemical markers in selected epithelial oncocytic tumors.

	CK7	CD117	Cathepsin K	CK20
RO	−	+	−	−
ChrRCC, eos	+	+	−	−
EVT	−/+	+	+	−
LOT	+	−	−	−
ESC-RCC	−	−	+	+

ChrRCC, eos: chromophobe renal cell carcinoma, eosinophilic variant; ESC-RCC: eosinophilic solid and cystic renal cell carcinoma; EVT: eosinophilic vacuolated tumor; LOT: low-grade oncocytic tumor; RO: renal oncocytoma.

**Table 4 cancers-16-01856-t004:** Diagnostic markers in different types of renal tumors.

Biomarker	Staining Pattern	Main Expression
CAIX	membranous/cytoplasmic	CCRCC, CCPRCT
CK7	cytoplasmic	PRCC, ChrRCC, CDC, CCRPCT, MTSCC, LOT
CK20	cytoplasmic	ESC-RCC
34βE12	cytoplasmic	CDC, RMC, CCPRCT
CD10	membranous	CRCC, PRCC, CDC
CD117	membranous/cytoplasmic	ChrRCC, RO, EVT
cathepsin K	cytoplasmic	EVT, ESC-RCC, AML

AML: angiomyolipoma; CCPRCT: clear cell papillary renal cell tumor; CCRCC: clear cell renal cell carcinoma; ChrRCC: chromophobe renal cell carcinoma; CDC: collecting duct carcinoma; ESC-RCC: eosinophilic solid and cystic renal cell carcinoma; EVT: eosinophilic vacuolate tumor; LOT: low-grade oncocytic tumor; MTSCC: mucinous tubular and spindle cell carcinoma; PRCC: papillary renal cell carcinoma; RMC: renal medullary carcinoma; RO: renal oncocytoma.

## Data Availability

Not applicable.

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
