# Peer review of "Diagnostic Biomarkers in Renal Cell Tumors According to the Latest WHO Classification: A Focus on Selected New Entities"

_cancers, 2024, doi:10.3390/cancers16101856_

Round 1

Reviewer 1 Report

Comments and Suggestions for Authors

This review entitled, "Diagnostic Biomarkers in Renal Cell Tumors According to the Latest WHO Classification: Focus on New Entities," provides an in-depth account of changes to renal tumor classification under WHO guidelines, with special attention paid to new entities that were identified through immunohistochemical profiling of immunohistochemical profiles for diagnosing new renal tumors - commendable given its depth and clinical utility; however, several areas in which it could be enhanced to better serve its intended audience:

Comments:

1. Clarity and Accessibility: This article contains many specialized terms, complex descriptions and language that might be difficult for those unfamiliar with its field to understand. Simplifying language or providing a glossary with less commonly used terms might make the piece more accessible to a wider audience.

2. Comparative Analysis: Although this review offers information on immunophenotypic profiles of new entities, more explicit comparison with prior classifications would help highlight what has changed or remains a challenge in diagnosing them.

3. Graphical Representations: Visual aids such as diagrams, flowcharts and tables summarizing key markers and their diagnostic significance can greatly enhance understanding. Visuals aids can be particularly effective at dissecting complex information so this could prove highly advantageous here.

4. Case Studies or Clinical Correlations: For added depth in reviewing diagnostic criteria and their effects, case studies or more explicit clinical correlations could enhance this review. They would allow a clearer illustration of their practical implications as they apply to patient care management.

5. Future Directions: Once reviewing current markers, this section could include potential future research directions or emerging markers that are currently under study to keep readers up-to-date with cutting edge research and collaboration in their fields of interest. It will not only inform them on cutting-edge issues but also inspire further study and collaboration efforts in that arena.

1. Methodology Section: Include a methodology section which details how the literature review and selection were undertaken for this article, to enhance scientific rigor while giving readers the chance to assess whether their analysis has covered every relevant source. This would enable readers to properly judge your efforts at compiling a comprehensive literature review.

2. Interdisciplinary Approach: Due to the complex interactions between genetic and molecular aspects discussed, an interdisciplinary approach could prove immensely valuable in this article. Consulting molecular pathologists or geneticists could give more insight into how biomarkers correlate with genetic changes within tumors.

3. Patient Outcome Data: Wherever possible, add more patient outcomes data derived from the new classification system to demonstrate its clinical utility in predicting prognosis and guiding therapy.

4. Expert Commentary: Involve a well-recognized expert in the field in providing additional insight on how a new classification might impact clinical practice and acceptability by the community.

5. International Perspective: Analyzing how these classifications are adopted or perceived across international settings could further make the review relevant to an international audience, considering differences in healthcare resources and practices across regions.

Overall, this review makes a substantial contribution to renal pathology by providing insight into WHO classification of renal tumors. With some modifications suggested for its enhancement, it could become even more helpful as an aid for both pathologists and clinicians alike.

Comments on the Quality of English Language

This review, "Diagnostic Biomarkers in Renal Cell Tumors According to the Most Up-to-Date WHO Classification: Focus on Certain New Entities", is superbly written with high academic English language standards that make this document suitable for scientific audiences with specific specialized needs.

Reviewer 2 Report

Comments and Suggestions for Authors

Compliments to the authors for a wonderful compilation of the pathological and immune histochemical details of few common subtypes of RCC.

What is the significance of the colored equal-to or not-equal-to signs in table 2?

Author Response

Thank you for your comment, there were some typos in Table 2, so we changed it accordingly.